# Effects of water, energy, and food security on household well-being

**Foster Awindolla Asaki** (ORCID) *, **Eric Fosu Oteng-Abayie, Franklin Bedakiyiba Baajike**

Department of Economics, Kwame Nkrumah University of Science and Technology, Kumasi, Ghana

* asakifoster@gmail.com

**Data Availability Statement:** The data that support the findings of this study is publicly available and have been added as a supplementary file.

**Funding:** The authors received no specific funding for this work.

## Abstract

Water, energy, and food insecurity are significant challenges that affect both economies and households, particularly in developing countries. These resources have an effect on households wellbeing, businesses, and all sectors of the economy, making them critical to ensuring household well-being, which is frequently measured by quintile welfare. As a result, there has been a significant increase in interest in securitizing these resources in order to mitigate their negative effects on household's wellbeing This study provides an empirical investigation of the determinants of water, energy, and food (WEF) security and the effect of water, energy, and food security on household well-being in Ghana. This study provides an empirical investigation of the determinants of water, energy, and food (WEF) security and the effect of water, energy, and food security on household well-being in Ghana. The study used a sample of 2,735 households from the Ghana Living Standard Survey (GLSS) Wave 7. We applied an instrumental variable probit, complementary log-log and ordered Probit estimation techniques for analysis. Empirical analysis reveals several important findings. Firstly, factors such as age, credit access, household location, employment status, and livestock ownership positively contribute to household water security, while remittances, water supply management, water bills, and water quantity have negative impacts. Secondly, age, marital status, household size, remittances, and livestock ownership significantly influence household energy security. Thirdly, marital status, household income, credit access, and household size are crucial determinants of household food security, with residence and region of household location exerting negative effects. Additionally, while water and energy security have a relatively lower impact on household well-being, food security emerges as a key driver in promoting household wellbeing. The study recommends that policymakers and stakeholders design and implement robust programs and interventions to sustain households' water, energy, and food supply.

## 1. Introduction

Water, energy, and food (WEF) resources continue to be critical for meeting basic human needs, addressing global concerns about hunger, health, and economic growth, and promoting long-term development [1–3]. Households rely on limited resources to survive and thrive [4,5].

**Competing interests:** The authors have declared that no competing interests exist.

Global water, energy, and food demand is outpacing supply [6,7], causing global scarcity. International predictions show an average of 40% increase in water, energy, and food demand if no action is taken [7,8]. According to Parry [9], satisfying global human needs in the next 20 years will require 50% more food, 50% more energy, and 30% more water. Unquestionably, the rise in demand for water, energy, and food resources and the general increase in resource prices indicate a future shortage [10,11].

Similarly, shortages of water, energy, and food demand in Sub-Saharan Africa (SSA) have worldwide consequences. For example, 40% of the SSA population lacks access to clean water [12]. They rely on contaminated surface water from rivers, streams, and lakes. In addition, 905 million people in SSA lacked access to clean cooking energy in 2018, with 848 million reliant on traditional biomass use [13,14]. According to the Food and Agriculture Organization (FAO) [15], Sub-Saharan Africa (SSA) accounted for one-third of the global population of 821 million people experiencing undernourishment, with a significant portion also facing micronutrient deficiencies [14]

Ghana is one of the countries in Africa privileged to have water, energy, and food resources available [16]. Still, the growing insecurity of these resources among the populace is worrying. Despite the various policies implemented by different governments in Ghana, it has been estimated that about 10.4%, 16%, and 11.9% of the population are still water, energy, and food insecure [13,17,18]. This indicates that water, energy, and food insecurity are prevalent in Ghana and thus important issues of policy concern in Ghana.

In Ghana, there are several major problems related to the determinants of water, energy, and food security. These challenges have significant implications for the country's sustainable development and the well-being of its population. For instance, Ghana faces water scarcity issues and quality issues due to pollution from industrial activities and inadequate sanitation systems affecting water sources due to inadequate infrastructure, population growth, and climate change impacts [19] Furthermore, A large portion of Ghana's population lacks access to reliable and affordable energy services coupled with insufficient electricity generation capacity, outdated infrastructure, and financial constraints contributing to energy access challenges [20]. Moreover, Ghana experiences food insecurity due to factors such as low agricultural productivity, post-harvest losses, and climate change impacts, inadequate irrigation systems, limited access to credit, and insufficient agricultural extension services also hinder food production and distribution [21]. However, identifying the determinants of water, energy, and food security, Ghana can work towards sustainable development, efficient resource allocation, and effective policy planning, thereby addressing the challenges and improving the well-being of its population.

Consequently, research shows that water, energy, and food security have significant impacts on household well-being, and the gaps in these areas can lead to negative outcomes. Specifically, Lack of access to clean and safe drinking water has been linked to health problems such as diarrhoea, cholera, and typhoid fever. Studies have also shown that women and children in households without access to clean water spend more time collecting water, which can negatively affect their education and economic opportunities [22]. Also, energy poverty, or the lack of access to reliable and affordable energy, can impact household well-being in several ways [23]. Without electricity, households are unable to refrigerate food or preserve medicine, and may have limited access to communication technologies that are necessary for work and education. Additionally, households may use inefficient and harmful energy sources such as kerosene or firewood, which can contribute to indoor air pollution and respiratory health problems. Moreover, food insecurity, or the inability to access nutritious and sufficient food, has been linked to a range of negative outcomes, including poor nutrition, physical and mental health problems, and reduced economic and educational opportunities. Studies have shown

that households without sufficient access to food are more likely to experience poverty and social exclusion [24,25].

Empirically, while some studies [26,27] have examined the impact of socioeconomic factors on water, energy, and food security, there is a need for more comprehensive investigations. Further research is required to consider household income, education, gender dynamics, and other relevant socio-economic variables to better understand the determinants and potential interventions [28,29]. Also, the effects of climate change on water, energy, and food security in Ghana are more pronounced. However, there is limited research on household-level adaptation strategies to mitigate the impacts of climate change. Therefore, expanding the determinants of water, energy, and food security will enable households to appreciate and adopt household level adaptative strategies such as water storage, renewable energy technologies, and resilient farming techniques that furthers reduces the insecurity of the resources understudy [20,28]. Furthermore, primary data gathered in particular parts of the nation has been the foundation for most research on food, energy, and water resources. Including national-level data representing households is crucial to achieve wider applicability and inform national policies.

Also, although some studies have identified the impact of interventions targeting Water, Energy, and Food Security on wellbeing, there is a need for research that explores context-specific interventions tailored to the Ghanaian context. This will contribute to the development of effective policies and programs that address the specific challenges faced by households in Ghana [30,31]. Furthermore, wellbeing is a complex and multi-faceted concept, and there is a need for standardized indicators and measurement tools specific to the Ghanaian context. Thus, an appropriate measure enhances the comparability of studies and facilitate a better understanding of the relationship between Water, Energy, and Food Security and wellbeing [29,32]. Addressing these research gaps will contribute to a deeper understanding of the effect of Water, Energy, and Food Security on wellbeing at the household level in Ghana and inform the development of targeted policies and interventions.

This study contributes to the existing empirical literature by examining the determinants of water, energy, and food security independently at the national level, in the context of Ghana, using a nationally representative dataset from the Ghana Statistical Services. The results guide policy direction to improve the factors that influence the security of water, energy, and food resources. Second, we further extend the empirical literature by independently examining the impact of water, energy, and food security on household well-being in Ghana. This will guide policymakers and stakeholders to factor water, energy, and food security improvement strategies or interventions into household well-being improvement programmes. To our knowledge, this is the first such empirical study on Ghana.

## 2. Literature review

### 2.1 Determinants of water, energy, and food security

The argument about water, energy, and food security is solely driven by increased demand for water, energy, and food resources. Empirically, water, energy, and food demand are rising, but the resources required to produce them are diminishing [33,34]. UNDP [35] and WHO/UNICEF [36] define water security as safe drinking water and sanitation. From the available literature, possible determinants of water security include household characteristics such as age, gender, education, marital status, and location (rural/urban). Other factors such as credit access, distance to a water source (or water bill), and quantity of water supply (or water bill) are found to predict water security [37–39].

The FAO defines food security as universal physical and economic access to adequate, safe, and nutritious food that meets daily needs and food preferences for an active and healthy life. Food security includes national food availability, household food access, utilization, and stability. Food is the most basic requirement for human survival. Nevertheless, variables affecting food security vary from country to country and from one locality to the other. Household characteristics like size, gender, educational attainment, marital status, and age of the household head; economic factors like nonfarm income, asset ownership, and input prices; and infrastructure factors like access to credit, market, road, and extension services are common determinants of food security highlighted in the literature [40–42].

Moreover, energy security is conceived as a complex multidimensional phenomenon that defies simple definition. Several researchers, however, attempt to provide diverging definitions of energy security. However, each definition recognizes that energy security revolves around the reliability and affordability of clean and modern energy services, which are essential for promoting human well-being [43–45]. According to Wang et al. [46], energy security depends on the availability, accessibility, affordability, and acceptability of modern and clean energy. Energy security determinants identified in the literature include household income, household size, employment status, electricity affordability, age of the household head, credit access, energy source, marital status, location such as rural or urban, and remittance [46–48].

## 2.2 Household well-being

The debate on well-being revolves around measurements in the empirical literature. Single indicators such as per capita income have been widely used to measure well-being. However, recent literature has redefined well-being as a multidimensional concept [49,50]. Household well-being is a multidimensional concept that captures the living conditions of the household with a focus on income, consumption, housing, productivity, health, education, leisure and social connections, economic and physical security, environmental quality, governance, and life expectancy [51]. The empirical findings on well-being as a multidimensional concept are based on the fact that single-indicator measures fail to capture overall well-being at the individual, household, and national levels [52]. In Ghana, a household's well-being, which is proxied by quintile welfare, is computed by the Ghana Living Standard Survey (GLSS 7). The quintile welfare is ranked from 1 to 5, with 1 being the lowest welfare and 5 being the highest level of welfare. The well-being as proxied by the quintile welfare of the household is influenced by several factors, including water, energy, and food security, as they are necessary to ensure the survival of the livelihood (2). One advantage of the quintile welfare is that it gives a clear picture of the households' characteristics and how they change when there is an increase or decrease in welfare [53].

## 2.3 Review of related literature

Theoretically, water, energy, food security and household well-being are hinged on the capability theory of human well-being [49]. The theory views capabilities as a set of alternative combinations of things an individual can achieve if the opportunity and freedom are given to them. Sen's theory also highlights that an individual has different preferences and desires. Thus, everyone can attain the same level of well-being given the same set of capabilities and freedom [49]

The method was developed in response to traditional welfarist notions of happiness, utility, and poverty being quantified by income and resources. Day et al. [54] claim that the standard approach is insufficiently broad to encompass and quantify the complete and expansive idea of well-being. However, income or resources are often not good proxies for measuring these

capabilities since the different levels of those resources or income may be required by different individuals to achieve the same capabilities. According to Robeyns [55], a capability-based approach does not negate resource utilization in well-being analysis or its implications. Sen [49] claims that talents are context-dependent. The agreed-upon minimum criteria are physical health, safety, nutrition, education, and social well-being. The capability hypothesis now encompasses water, energy, and food security, as well as their relevance to well-being.

However, the capability method is also used to establish a robust correlation between hydro-social ties and human well-being [56,57]. Water scarcity was seen in terms of hydro-social interactions and human well-being. Water reproduction is an interaction between hydro-social relations and human well-being. Water reproduction involves survival and livelihood. So, safe water, among other things, enables people to flourish.

The capability method was also utilized to examine the link between energy security and human welfare. The capabilities method characterized energy inside the model by splitting well-being into groups and studying each level's interaction. The basic capabilities or functions include energy supply, energy consumption, and energy services such as lighting, cooking, heating, and cooling [54]. The model also included secondary capabilities such as preparing food, reading and accessing the internet. Intuitively, households' ability to afford and access clean and modern energy types, energy supply, and energy services improves their well-being.

The capabilities concept extends to food security challenges such as hunger, malnutrition, and famines, stating that human capacities determine the right to goods [58]. The capability approach separates means from ends, avoiding malnutrition and hunger. The FAO's [59] food security components of availability, access, and use are addressed through the capability approach. Thus, ensuring household food availability and access will increase their well-being. Finally, the capabilities approach to well-being indicates that well-being is a multidimensional approach that heavily relies on water, energy, and food security. According to the hypothesis, water, energy, and food-secured households do better at enhancing their well-being.

Empirically, studies on the determinants of food security conclude that income, age, gender, education, and nonfarm income increase food security while household size and high dependency ratio reduce it [60,61]. In Ghana, case studies have been used to study food security determinants. The results showed that farm size, off-farm, marital status, and credit availability improve food security, whereas household size, rural housing, age, and land size lower it [27,62]. Case studies lack scientific rigour and offer little support for extrapolating the findings to the national level. Case studies are also difficult to replicate, and the results may be skewed due to the researchers' subjective feelings about the choice of samples under study [63]. Therefore, it is imperative to explore the determinants of food security using a dataset representative of the entire populace to allow for scientific generalization and reliable policymaking.

Previous research [64,65] found the age of household heads and farmers, income level, farm location, and rural-urban household location as factors that promote water access and security. Conflicts and nearness to water canals worked against water security [65]. Water security research in Ghana has focused on households' water access [66–71], water security and water demand [26], water security on agriculture activities [72], and the socio-economic determinants of sources of drinking water [73]. However, the determinants of household water security have not been explored in the empirical literature. One research gap on water security in Ghana at household levels is the lack of understanding of the sociocultural factors that influence water use behavior among households [74]. Further research is needed to examine the socio-economic implications of water insecurity at the household level in Ghana.

Concerning energy security, the dominating themes in the literature include renewable energy development [75,76], determinants of energy supply security [77], and energy security

and sustainability [78]. Except for Ankrah and Lin [75], there is a gap in research on the drivers of energy security among Ghanaian families. In terms of the drivers of energy security, Ningi, Taruvinga & Zhou [79] and Nagothu [80] concluded that marital status, energy availability, and income sources improve energy security. Rural and low-income households have also been more energy-inefficient than urban and high-income households [79]. Thus, it can be concluded that access to energy services varies by location and that more households lack clean cooking facilities in developing countries such as Ghana. One research gap on energy security in Ghana at household levels is the lack of empirical evidence on the impacts of energy poverty on human development outcomes. Despite the recognition of the importance of energy security for sustainable development, little research exists on how energy security affects livelihood outcomes at the household level in Ghana [81].

## 3. Methodology and model specification

### 3.1 Theoretical model

The theoretical framework is partitioned into two parts: the impact of water, energy, and food security on household well-being and, secondly, the determinants of water, energy, and food security. In analysing the impact of water, energy, and food security on household well-being, the study relies on the capability theory [49]. The theory considers water, energy, and food security essential for improving households' well-being. The relationship between household well-being and water, energy, and food security following the household well-being specification of Sen [49], Nagothu [80], Kimengsi et al. [82] is expressed in Eq (1):

$$S_i = f(WS, \ ES, \ FS, \ X) \tag{1}$$

Where household well-being ($S_i$), WS represents Water Security, ES denotes Energy security, FS represents Food Security, and X represents other factors that affect household well-being. The household well-being ($S_i$) specification is expanded to include other factors as shown in Eq (2):

$$S_i = f(WS, \ ES \ FS, \ age, \ gender, HS, \ MS, \ REM, \ CA, \ TW, \ HY, RG, RS) \tag{2}$$

Where WS represents water security, ES denotes energy security, food security, HS denotes the household size, MS represents marital status, REM is remittance, CA represents credit access, TW represents total wage income, HY is household income, RG represents the region of the household and RS is the residence of the household.

Apart from the relationship between household well-being and water, energy, and food security, the study also looks at the factors influencing water, energy, and food security. This study assumes that household characteristics and sector-specific factors affect water, energy, and food security, as represented in Eqs (3) to (5).

$$FS = f(AGE, \ MS, \ GENDER, \ HY, \ HS, \ ED, \ CA, \ RS, \ RG, \ LSO, \ EMS) \tag{3}$$

$$ES = f(AGE, \ MS, \ GENDER, \ HY, \ HS, \ CA, \ RS, \ RG, \ ENS, \ EB, \ NFY, \ RM, \ EMS) \tag{4}$$

$$WS = f(AGE, \ MS, \ GENDER, \ HY, \ HS, \ CA, \ RS, \ RG, \ WSM, \ WB, \ WSC, \ WD, \ WQ) \tag{5}$$

Where FS, ES, and WS represent food, energy, and water security, respectively. HY is household income. HS denotes the household size. CA represents credit access. RS is the residence. RG means region. WSM denotes water supply management. WB represents the water bill. WSC implies water supply management. WD represents the distance to a water source. WQ

represents water quantity. EMS represents employment status. Moreover, ED represents educational status and LSO denotes livestock ownership. ENS represents energy sources; EB implies an electricity bill; NFY is the nonfarm income; RM means remittance.

## 3.2 Econometric model specifications

In analyzing the determinants of water, energy, and food security, this study follows Penman and Johnson [83] to specify the empirical model using the complementary log-log model (CLL). As is always the case with linear regression models, the dependent variable $Y$ (water, energy, and food security) is modelled against a set of linear predictors. $X_1, X_2, X_3. \ldots \ldots..X_{k-1}$ where $X_i$ ($i = 1, 2, 3, \ldots... k - 1$) denotes predictor variables to be adjusted for. When $Y_i$ ($i = 1, 2, 3 \ldots n$) are independent, identically distributed Bernoulli random variables. The expected value is the fraction of positive ($Y = 1$) responses, $\pi$, also known as the predicted probability. For the general case with $k - 1$ predictor variables with no interaction terms, the CLL function $log\{-\log(1 - \pi_i)\}$, maps the range of predicted probabilities, $\pi_i$, onto the real line ranging from $-\infty$ to $\infty$.

$$log\{-\log(1 - \pi_i)\} = \left(\beta_0 + \beta_1 X_{i,1} + \beta_2 X_{i,2} + \cdots + \beta_{k-1} X_{i,\gamma-1}\right) \tag{6}$$

Eq 2 can also be denoted in a matrix form as given in Eq (7):

$$log\{-\log(1 - \pi_i)\} = X^T \beta \tag{7}$$

Where $\beta_i$ the coefficients to be estimated are, $X$ is a $k \times 1$ column vector of beta coefficients ($\beta = \beta_0, \beta_1, \ldots... \beta_{k-1}$) and $\beta_0$ is the intercept. Maximum likelihood estimation is used to determine the parameters. The water, energy, and food security models are empirically specified as Eqs (8) to (10).

$$WS = \beta_0 + \beta_0 age_{i1} + \beta_2 Ms_{i2} + \beta_3 gender_{i3} + \beta_4 HY_{i4}$$
$$+\beta_5 HS_{i5} + \beta_6 CA_{i6} + \beta_7 RS_{i7} + \beta_8 RG_{i8} + \beta_9 WSM_{i9} \tag{8}$$
$$+\beta_{10} WB_{i10} + \beta_{11} WSC_{i11} + \beta_{12} WD_{i12} + \beta_{13} WQ_{i13} + \varepsilon$$

$$ES = \theta_0 + \theta_0 age_{i1} + \theta_2 Ms_{i2} + \theta_3 gender_{i3} + \theta_4 HY_{i4}$$
$$+\theta_5 HS_{i5} + \theta_6 CA_{i6} + \theta_7 RS_{i7} + \theta_8 RG_{i8} + \theta_9 ENS_{i9} \tag{9}$$
$$+\theta_{10} EB_{i10} + \theta_{11} NFY_{i11} + \theta_{12} RM_{i12} + \theta_{13} EMS_{i13} + \varepsilon$$

$$FS = \emptyset_0 + \emptyset_0 age_{i1} + \emptyset_2 Ms_{i2} + \emptyset_3 gender_{i3}$$
$$+\emptyset_4 HY_{i4} + \emptyset_5 ED_{i5} + \emptyset_6 CA_{i6} + \emptyset_7 RS_{i7} \tag{10}$$
$$+\emptyset_8 RG_{i8} + \emptyset_9 LSO_{i9} + \emptyset_{10} EMS_{i10} + \varepsilon$$

Where all variables are already defined. Moreover, $\beta$, $\theta$ and $\emptyset$ are the coefficient to be estimated and $\varepsilon$ is the error term.

Following Cameron and Trivedi [84], the study employed ordered choice specifications to examine the effect of water, energy, and food security on the well-being of households in Ghana, as specified in Eq (11).

$$y* = x'\beta + \varepsilon \tag{11}$$

Where $y$ is the observed outcome and $y^*$ is the underlying continuous unobservable or latent dependent variable, $x$ is a vector of explanatory variable and $\varepsilon$ is the error term.

 

Following the specification in Eq (11) where $y$ is a continuous and unobservable latent measure of well-being and $x$ is a vector of explanatory variables that influence well-being, $\beta$ is the coefficient to be estimated and $\varepsilon$ is the error term which is normally distributed. The well-being model, is empirically specified as Eq (12):

$$S_i = \alpha_0 + \alpha_{i1}WS_{i1} + \alpha_{i2}ES_{i2} + \alpha_{i3}FS_{i3} + \alpha_4 age_{i4} + \alpha_5 Ms_{i5}$$
$$+\alpha_6 gender_{i6} + \alpha_7 HY_{i7} + \alpha_8 HS_{i8} + \alpha_9 CA_{i9} + \alpha_{10}RS_{i10} \qquad (12)$$
$$+\alpha_{11}RG_{i11} + \alpha_{12}TW_{i12} + \alpha_{i13}gender_{i13} + RM_{i12} + \epsilon$$

Where $S_i$ represents household well-being. All other variables already defined. Also, $\alpha_i$ are the coefficient to be estimated and $\varepsilon$ is the error term.

Again, the well-being indicators are coded into groups whereas $y$ can be observed as given

$$y = \begin{cases} 0 & if\ y \leq 0,\ for\ lowest\ wellbeing \\ 1 & 0 < y \leq \mu_1 \\ 2 & \mu_1 < y \leq \mu_2 \\ 3 & \mu_1 < y \leq \mu_3,\ highest\ wellbeing \end{cases} \qquad (13)$$

Where $\mu's$ are the unknown parameters to be estimated with $\beta$. After converting the error term's mean and variance to 0 and 1, the probabilities associated with the coded dependent variables are expressed as:

$$\Pr(y = 0/x) = \emptyset(x'\beta)$$
$$\Pr\left(y = \frac{1}{x}\right) = \emptyset(\mu_1 - x'\beta) - \emptyset(x'\beta)$$
$$\Pr\left(y = \frac{2}{x}\right) = \emptyset(\mu_2 - x'\beta) - \emptyset(\mu_1 - x'\beta) \qquad (14)$$
$$\Pr\left(y = \frac{3}{x}\right) = 1 - \emptyset(x'\beta)$$

For M alternatives, where $y = j$ if $\mu_{j-1} < y < \mu_j$ and $j = 1, 2\ldots.m$. $\mu_j = -\infty\ and\ \mu_m = \infty$, then

$$\Pr(y_i = j) = \Pr(\mu_{j-1} < y \leq \mu_j)$$
$$= \Pr(\mu_{j-1} < x'\beta + \varepsilon \leq \mu_j)$$
$$= \Pr(\mu_{j-1} - x'\beta < \varepsilon_i \leq \mu_j - x'\beta) \qquad (15)$$
$$= F(\mu_j - x'\beta) - F\left(\Pr(\mu_{j-1} - x'\beta)\right)$$

Where F is the cumulative distribution function of $\varepsilon_i$ and the regression parameter is $\beta$ and $m - 1$ are the threshold parameters of $\mu_1 \ldots \mu_{m-1}$ which are obtained by maximizing the log-likelihood with $P_{ij} = \Pr(y_i = j)$.

For positive coefficients of the estimation,

$$0 < \mu_1 < \mu_2 < \mu_3 \qquad (16)$$

The coefficients estimated in the model cannot directly be interpreted, thus the marginal effects of the model will be estimated wherein a change in one independent variable will result in a change in the distribution of the outcome variable as given Eq (5).

 

The marginal effect (ME) on the probability of choosing alternative j when regression $x_\tau$ changes are given as

$$\frac{\partial \mathrm{Pr}(y_i = j)}{\partial x_\tau} = (\mathrm{F}'(\mu_{j-1} - x'\beta) - \mathrm{F}'(\mu_j - x'\beta))\beta_\tau \tag{17}$$

Where from $i$ = 1, 2, 3, 4 *and* 5 denotes the groups of household well-being levels, x is the explanatory variable $\mu's$ represents the cut-off value for the ordered probit [84].

## 3.3 Estimation techniques

The water, energy, and food security indices were built using principal component analysis (PCA). The PCA turns original variables into new uncorrelated variables called primary components. By analyzing the data in principal component form, Helena et al. [85] claim to gain extensive insight into the factors that determine the complete dataset. Sarbu and Pop [86] stated that the PCA gives an objective approach to producing indices of this type that can account for the variation in the dataset concisely.

The study examines the determinants of water, energy, and food security using the complementary log-log. The available binary responses include logit, probit, and complementary log-log models. The choice of these models depends on whether the distribution of data is symmetric or asymmetric. The Probit and logit models are the best choices when the distribution of the data is symmetric, whereas the complementary log-log model provides satisfying results when the data is asymmetric in distribution [87]. Therefore, the study adopted the complementary log-log model because of the asymmetric distribution of energy security, and food security data and instrumental variable probit for estimating the water security model due to the possibility of endogeneity problem.

The study also looked at how water, energy, and food security affect household well-being. Following Greene [88] and Mallick and Rafi [89], the study used an ordered Probit model to analyze household well-being (ordered from 1 to 5). To assess the robustness of the ordered Probit results, the model's projected probabilities were compared to the dependent variable's actual means.

## 3.4 Data source

The study employed data from the Ghana Statistical Service's 7th Ghana Living Standard Survey (GLSS 7). The Ghana Statistical Service (GSS) conducts the nationally representative household survey known as the Ghana Living Standards Survey (GLSS). Through the collection of data on demographics, education, health, employment, migration, housing, income, spending, agriculture, and data protection, it assesses the standard of living and general well-being of the people in Ghana. In order to capture seasonal variations in consumption and production, the survey spans a full year and includes approximately 15,000 households in 1,000 Enumeration areas throughout the nation [90]. Food, energy, water, fuel consumption, household agriculture, access to financial services, and asset ownership are among the variables included in the survey [90]. The data covers 14,009 households nationwide. This study drew a sample of 2,735 families since they provided complete data on food, energy, water, and well-being, which are the study's major variables. The detail of the variables is shown in Table 1 below.

The Ghana Living Standard Survey (GLSS) is a nationally representative dataset collected by the Ghana Statistical Service with the support of the World Bank. Empirically, several studies have used the GLSS for household studies [91–95]. As an extension to the existing literature, this study focuses on the food, energy, and water resource sectors.

**Table 1. Data variables.**

| Variables | Description | Unit of measurement |
|---|---|---|
| WS | Water Security. Household access to safe drinking water and sanitation. | dummy where 1 = high secure and 0 = least secured |
| ES | Energy Security.<br>Availability, accessibility, affordability, and acceptability of modern and clean energy services | dummy where 1 = high secure and 0 = least secure |
| FS | Food Security.<br>physical and economic access to sufficient, safe and nutritious food that meets the diary needs and food preference for an active and healthy life | dummy where 1 = high secure and 0 = least secure |
| LSO | Livestock ownership.<br>Concern livestock owned or fish farming activities undertaken by households. | Household heads with livestock |
| EMS | Employment Status.<br>The status of the household head whether employed or unemployed. | Dummy where 1 = employed and 0 = otherwise |
| ED | Educational Status.<br>The level of education of the household beginning from no education to highest level of learning. | Higher level of education of the household head |
| CA | Credit Access.<br>This concerns on loans contracted or negotiated by the household in terms of money or goods. | Dummy where 1 = heads with access and 0 = otherwise |
| RG | Region.<br>The region households are located whether southern part of Ghana or the northern part of Ghana. | Dummy where 1 = households at south and 0 = households at north |
| ENS | Energy Source.<br>Households source of energy includes National grid, solar etc. | Source of power the household |
| EB | Electricity Bill. Amount of money (GHC) spent in paying electricity bill or buying prepaid. | Amount pay or spend regular on electricity |
| NFY | Nonfarm Income. Nonfarm income are all incomes household received which are not generated from farming activities or Net income from nonfarm enterprise | Income from nonfarm Enterprises |
| RM | Remittance. Remittances are regular or irregular contributions in terms of money, goods and food made to or received from person(s) living abroad or elsewhere. | Income from remittances |
| TG | Total Wage.<br>Total household wage income received. | Total household wage |
| WSM | Water Supply Management.<br>Water supply system operation and management. | . Dummy where 1 = public supply management and 0 = private supply management system |
| WB | Water Bill. Amount of money (GHC) households pays regularly for usage of water. | Amount paid regularly as bill for the usage of the water |
| WSC | Water Supply Consistency. Frequency of water sources availability for household's access. | Regular availability of the source of the water supply |
| WD | Water Distance.<br>The distance between households and water sources. | Distance of water sources from dwelling measured in meters. |
| WQ | Water Quantity.<br>The quantity of water household's uses per day for their activities and consumption. | Quantity of water household uses per day |
| HS | Household Size. Number of people residing in each household. | Number of people in a household |
| RS | Residence. Location of the household (urban area or rural area) | Dummy where 1 = Urban and 0 = Rural area |

*(Continued)*

**Table 1.** (Continued)

| Variables | Description | Unit of measurement |
|---|---|---|
| MS | Marital Status. Households with or without married couple. | Dummy where 1 = Married and 0 = Unmarried |
| HY | Household income. Total Household net Income | Total household net income |

## 3.5 Measurement of variables

Following Sinyolo et al. [64], we used principal component analysis to create a water security index using variables from the GLSS7 data module related to access to safe drinking water for households and sanitation. The water security index is based on access to clean water, sanitation, and water storage. They had factor scores or eigenvalues exceeding 1. A household water security index was created using the components that captured the most variation (71%). The scores of the water security index ranged from -1.01 to 7.40. All scores that were below zero were considered as households with the least security and households with values above zero were also considered as households with high level of security. Thus, water security index was normalized to zero (0) indicating households that were ranked as least secured based on their scores and one (1) indicating households with high level of water security, which is required by the instrumental variable probit model for studying water security determinants. Normalizing an index into one (1) and (0) helps to remove the effect of variable scale, makes the index comparable with other indices, helps to standardize data, and ensures more accurate predictions [96].

Empirically, Ningi et al. [79] used the multidimensional energy poverty index to generate an energy security index for analysing the household energy security status. Following Ningi et al. [79], and Sadath and Acharya [95], three broad categories of energy use were identified at the household level, such as lighting, cooking, and additional measures. However, unlike the multidimensional approach for measuring energy security in developed countries, with different natures of energy consumption and socio-economic characteristics, adopting such an index fully for measuring energy security status in households in developing countries may result in misleading conclusions since there are disparities between developed and developing countries. Different from Ningi et al. [79] and Sadath and Acharya [95], the principal component analysis was adopted to generate an energy security index using the three broad categories and the other additional measures of energy use. The principal components retained, that is, with eigenvalues above one (1) for the creation of the energy security index, captured 65% of the variation. The index scores values ranged from -2.96 to 1.59. Moreover, by standardizing the index, the energy security index was normalized to zero (0) indicating households that least secured and one (1) indicating households with higher level of energy security and was used as the dependent variable in a complementary log-log model for examining the determinants of energy security.

Food consumption scores are commonly used across countries and take into account household dietary diversity and the nutritional value of food consumed. Again, principal component analysis was used to create a food security index based on food types such as grains, tubers and root crops, pulses, vegetables, fruits, meat and fish, oil, milk, and sugar. The index scores values ranged from -3.25 to 16.36. However, scores below zero were categorised households with least food security and scores above one (1) were regrouped as households with high level of food security. Thus, food security index was normalized to 0 and 1 and utilized as the dependent variable in a complimentary log-log model. Other variables included in the

study are the household head's age, gender, marital status, region of origin, area of residence (whether urban or rural), household size and income, credit access, educational status, quintile welfare as a proxy for household well-being, and other sector-specific variables related to water, energy, and food security.

## 4. Discussion of results

### 4.1 Descriptive analysis

The demographic characteristics of the households are presented in Table 2 below. The result indicates that the average age of a household head is approximately 48 years old, with a mean house size of 5 people. Furthermore, 0.86 present of the population lives in cities. Moreover, 0.62 percent of the population have access to credit and the mean income for each household was estimated to be GHC699.288 in a year.

### 4.2 Distribution of water, energy, and food security

Fig 1 illustrates the distribution of water, energy, and food security. From Fig 1, more households are less secure than more secure households. Again, the graph shows that more households are more energy secure as compared to households that are not. Similar to water security, more households are less food secure than households that are more households.

Table 3 shows the summary statistics for water, energy, and food security variables. From the table, we can see that the mean value for water security is 0.356, which suggests that the level of water security is relatively low. The standard deviation of 0.479 indicates that the level of water security varies widely across the sample.

The mean value for energy security, on the other hand, is higher at 0.601, indicating a higher level of energy security. The negative skewness of -0.413 suggests that the distribution is slightly skewed to the left, with most of the data distributed towards higher levels of energy security.

Finally, the mean value for food security is relatively low at 0.202, with a standard deviation of 0.402 indicating a wide variation in the levels of food security. However, the high positive skewness of 1.483 and kurtosis of 3.199 suggest that the distribution is highly skewed to the right and has a very peaked distribution, with most of the observations concentrated towards the lower end of the scale. Overall, the findings suggest that there is a need to improve water and food security, while energy security appears to be moderately satisfactory.

### 4.3 Determinants of water, energy, and food security

**4.3.1 Determinants of water security.** Table 4 presents instrumental variable probit output on the determinants of water security among households in Ghana.

The findings indicate that age, credit access, region of household, and livestock ownership are more likely to increase household water security. Tuong et al. [97] also found that older

**Table 2. Household demographic characteristics.**

| Variable | Obs | Mean | Std. Dev. | Min | Max |
|---|---|---|---|---|---|
| Age | 2,735 | 47.86472 | 14.79959 | 16 | 99 |
| Household Size | 2,735 | 5.006581 | 2.844281 | 1 | 24 |
| Urban | 2,735 | 0.858867 | .3482228 | 0 | 1 |
| Household income | 2,735 | 699.2881 | 162226.5 | -6529171 | 3364605 |
| Credit Access | 2,735 | 0.619378 | .4856285 | 0 | 1 |

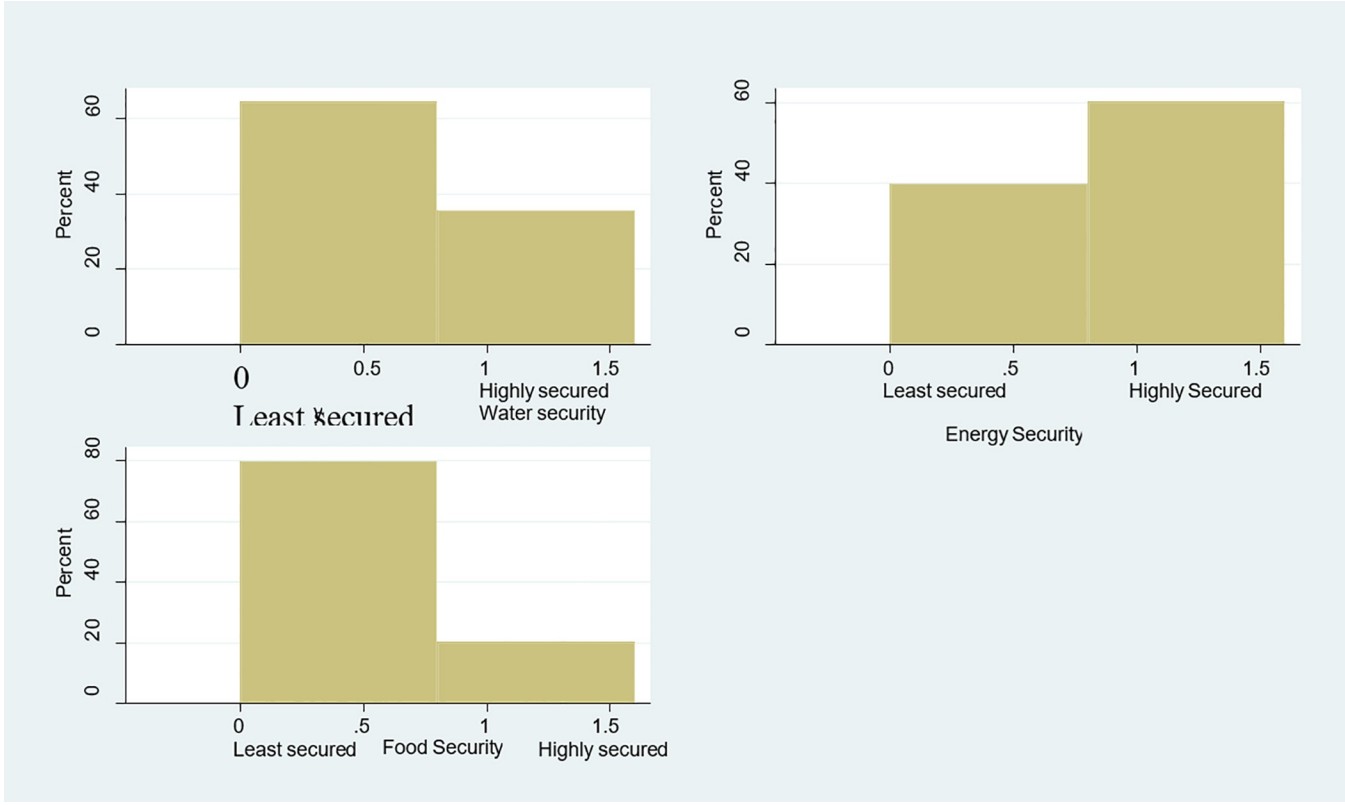

**Fig 1. Distribution of water, energy, and food security.**

**Table 3. Summary statistics of water, energy, and food security.**

| Variables | Mean | Std. Dev. | Variance | Skewness | Kurtosis |
|---|---|---|---|---|---|
| Water Security | 0.356 | 0.479 | 0.229 | 0.601 | 1.361 |
| Energy Security | 0.601 | 0.490 | 0.240 | -0.413 | 1.171 |
| Food Security | 0.202 | 0.402 | 0.161 | 1.483 | 3.199 |

**Table 4. Determinants of households water security in Ghana.**

| Variables | dy/dx | Std. Err. | z | P>z | [95% Conf. Interval] | |
|---|---|---|---|---|---|---|
| Age | 0.278 | 0.082 | 3.41 | 0.001 | 0.118 | 0.438 |
| Marital status | 0.000 | 0.077 | 0.01 | 0.996 | -0.150 | 0.150 |
| Gender | -0.063 | 0.070 | -0.89 | 0.371 | -0.201 | 0.075 |
| Household income | -0.020 | 0.019 | -1.08 | 0.282 | -0.057 | 0.016 |
| Remittance | -0.049 | 0.029 | -1.72 | 0.085 | -0.106 | 0.007 |
| Credit Access | 0.147 | 0.057 | 2.57 | 0.010 | 0.035 | 0.259 |
| Residence | 0.056 | 0.076 | 0.74 | 0.460 | -0.093 | 0.205 |
| Regions | 0.601 | 0.057 | 10.62 | 0.000 | 0.490 | 0.712 |
| Employment status | 0.178 | 0.178 | 1 | 0.317 | -0.171 | 0.527 |
| Livestock ownership | 0.135 | 0.068 | 1.99 | 0.047 | 0.002 | 0.268 |
| WSM | -0.458 | 0.085 | -5.37 | 0.000 | -0.625 | -0.291 |
| Water Bill | -0.497 | 0.060 | -8.23 | 0.000 | -0.616 | -0.379 |
| water_supply | -0.157 | 0.147 | -1.07 | 0.285 | -0.444 | 0.131 |
| water Quantity | -1.522 | 0.813 | -1.87 | 0.061 | -3.116 | 0.072 |

household heads are more likely to adopt water conservation practices and to implement measures to protect water resources. As a result, older household heads attain water security compared to younger household heads. Furthermore, credit access increases household purchasing power to enable them to invest water infrastructure and services. This allows for the constant supply of water to meet households water needs including cooking, and other domestic and economic activities. With credit, households can afford upfront costs, which increase their access to clean water sources while reducing reliance on unsafe or distant sources. The ability of the private sector to participate in water infrastructure is contingent upon financing availability, and this can lead to enhanced population access to water services [98].

Moreover, households located in the northern part of Ghana are generally more likely than those in the southern region to have safe access to water. Even though the southern sector enjoys two peaks of rainfall and are expected to be relatively more water secure, the opposite is, however observed. This discrepancy in water security between the two sectors may be due to the negative impacts of prevalent illegal mining activities, which are especially common in the southern part of Ghana. This result is consistent with the findings by Mensah et al. [99], who found that illicit mining has a negative impact on the economy, society, and environment in Ghana. It causes biodiversity loss, land degradation, water pollution, and lost mineral revenue. This affects the quantity and quality of water available for use in homes and farms. It also increases the cost of treating water, which is transferred to households, thereby making compromising water security due to cost. Similarly, Asamoah [100] found high concentrations of heavy metals in irrigation water, particularly zinc and manganese, which are harmful to consumers' and farmers' health; whiles Nukpezah [101] also noted a decline in the quantity and quality of water in mining communities, which affects the availability of potable water.

Livestock ownership improves household water security by requiring water supplies for animals, allowing for investments in water infrastructure, diversifying income for water-related expenses, and strengthening agricultural resistance to climate variability. Confirming expectations, Sinyolo et al. [64] also observed a positive relationship between livestock's ownership and water security. The results also revealed that households that depend on public Water supply management system (WSP) are less likely to be water secure, relative to households that patronize private WMS such as self-supply, NGO, among others. This shows the efficiency in the management and supply of clean water by the private individuals, while revealing the friction and inefficiencies associated with public goods and services.

On the other hand, water bills, and Water Quantity among households are more likely to decrease household water security.

In line with expectations, an increasing level of water bills tends to encourage households to turn to unsafe water sources due to their inability to afford the high water bills. Confirming the findings by United Nations World Water Development Report [102], water insecurity is rooted in institutional, social, and technological systems that shape water availability, access, and use. Water quantity plays a critical role in determining household water security. Surprisingly, the findings indicate that increasing the quantity of water available for household use may actually decrease overall water security. Howbeit counterintuitive, the finding suggests that household water security cannot be solely determined by the amount of water available, but rather must be complemented by other factors such as water quality.

**4.3.2 Determinants of household's energy security in Ghana.** Table 5 presents the complementary loglog output of the determinants of household energy security in Ghana.

Analytically, the findings are consistent with empirical findings, age, marital status, household size, and livestock ownership are more likely to increase household energy security. A study by Zhao et al. [103] found that the elderly was more likely to experience energy security in China. This implies that as household heads age, they become more aware of the importance

**Table 5. Determinants of household's energy security in Ghana.**

| Variables | Coef. | Std. Err. | z | P>z | [95% Conf. Interval] | |
|---|---|---|---|---|---|---|
| Educational status | 0.008 | 0.033 | 0.25 | 0.800 | -0.056 | 0.072 |
| Age | 0.179 | 0.055 | 3.24 | 0.001 | 0.071 | 0.287 |
| Marital status | 0.068 | 0.036 | 1.87 | 0.061 | -0.003 | 0.140 |
| Gender | -0.025 | 0.040 | -0.63 | 0.530 | -0.105 | 0.054 |
| Household income | -0.013 | 0.011 | -1.21 | 0.226 | -0.035 | 0.008 |
| Residence | -0.075 | 0.038 | -1.97 | 0.049 | -0.150 | 0.000 |
| Region | -0.044 | 0.036 | -1.24 | 0.215 | -0.114 | 0.026 |
| Household Size | 0.012 | 0.006 | 1.99 | 0.046 | 0.000 | 0.024 |
| Credit Access | 0.026 | 0.030 | 0.85 | 0.394 | -0.033 | 0.084 |
| Electricity Bill | -0.010 | 0.021 | -0.48 | 0.633 | -0.052 | 0.031 |
| wage income | 0.000 | 0.008 | 0.03 | 0.978 | -0.016 | 0.016 |
| Remittance | -0.053 | 0.016 | -3.26 | 0.001 | -0.084 | -0.021 |
| Employment status | 0.003 | 0.114 | 0.03 | 0.980 | -0.220 | 0.226 |
| Livestock ownership | 0.075 | 0.038 | 1.98 | 0.047 | 0.001 | 0.150 |
| Cons | 0.008 | 0.033 | 0.25 | 0.800 | -0.056 | 0.072 |

of energy security and, therefore, tend to invest in more reliable sources of energy. Again, marital status, particularly being married, is likely to improve household energy security through resource pooling, efficient energy management, and mutual support during difficult times, and access to larger social networks for assistance and information [79].

Larger households benefit from economies of scale in energy consumption through shared energy services and appliances, which lowers per capita energy demand and costs. As a result, each person uses less electricity, cooking fuel, and energy for lighting, heating, and cooling. Furthermore, larger households tend to have higher income and wealth, allowing for income diversification and investment in renewable energy. These factors work together to increase household energy security by lowering per capita energy consumption and costs [104].

Livestock ownership improves household energy security by providing income for modern energy services, food that reduces the need for energy-intensive cooking, manure for soil fertility and biogas production, and fuel from dung and crop residues, particularly in rural areas with few energy options [105].

Contrarily, residence and remittance were found to be more likely to decrease household energy security. Urban residences frequently have lower energy security due to population density, infrastructure, and industrial activity. This results in increased competition for resources and higher expenses. Renewable energy adoption is limited due to a lack of available space for production and storage. Urban residents rely significantly on centralised energy networks, rendering them susceptible to supply outages and price variations. Apartment-style living reduces individual control over energy consumption, increasing reliance on external sources [106,107].

Remittances, despite providing financial aid to households, unexpectedly diminish energy security. They alter consumption patterns, increasing energy demands for appliances, heating, and cooling systems, straining household budgets. Additionally, remittances may spur energy-intensive purchases, further elevating energy usage. Reliance on remittances may hinder investment in energy-saving devices or renewable energy systems. Moreover, remittance-dependent households are vulnerable to fluctuations in remittance inflows, impacting energy access and affordability during economic instability [108,109].

**4.3.3. Determinants of household food security.** Table 6 presents the complementary loglog findings on the determinants of food security among household in Ghana.

The findings indicate that marital status, household size, credit access, and household income are more likely to increase household food security.

Marital status can influence household food security via a variety of processes. Married households may have more income sources, negotiating power, social support, and access to food assistance programmes than single or divorced households [110]. In contrast, single or divorced households may face greater obstacles such as lower income, higher dependency ratio, higher food expenditure, and increased risk of food insecurity [111]. As a result, marital status especially been married is a significant predictor of household food security, particularly in urban areas where food prices are high and availability is unclear. Also, household size also plays a role in this association [110].

Food security and household size are positively connected, according to research by [61]. The reason could be that larger households may have surplus labor, which they can deploy for food preparation, processing, and storage [112]. They can also pool resources to buy a refrigerator or a freezer to preserve food for a long time. This would reduce food wastage and ensure that the household has enough food available throughout the year.

Access to credit improves household food security by allowing investments in agricultural inputs, technology, and irrigation systems, thereby increasing productivity. It provides financial flexibility during crop failures and income fluctuations, allowing for food purchases. Furthermore, credit enables income diversification beyond agriculture, which improves food security [113,114].

Household income is a critical factor in determining household food security. Higher income levels give families more resources to buy an adequate and nutritious diet, lowering the risk of hunger and malnutrition. Increased income allows households to access a broader range of food items, improving dietary diversity and nutritional intake. The findings align with prior research indicating a positive relationship between household income and food security [115,116]. Higher household income levels contribute to improved food security by enhancing families' ability to afford a diverse and nutritious diet, thereby reducing the risk of hunger and malnutrition.

Also, residence and region of a household are important factors that determine the extent of food security of a household. With regards to residence, Households in urban areas often experience lower levels of food security compared to those in rural areas. Urbanization

**Table 6. Determinants of household food security.**

| Variables | Coefficients | Std. Err. | z | P>z | [95% Conf. Interval] | |
|---|---|---|---|---|---|---|
| Educational status | 0.004 | 0.021 | 0.2 | 0.840 | -0.038 | 0.046 |
| Age | 0.025 | 0.036 | 0.69 | 0.490 | -0.045 | 0.094 |
| Marital status | 0.041 | 0.024 | 1.69 | 0.091 | -0.006 | 0.088 |
| Gender | 0.047 | 0.029 | 1.63 | 0.102 | -0.009 | 0.104 |
| Residence | -0.138 | 0.032 | -4.26 | 0.000 | -0.202 | -0.075 |
| Regions | -0.125 | 0.025 | -4.94 | 0.000 | -0.175 | -0.076 |
| Household Size | 0.014 | 0.004 | 3.83 | 0.000 | 0.007 | 0.021 |
| Credit Access | 0.068 | 0.021 | 3.23 | 0.001 | 0.027 | 0.110 |
| Household income | 0.032 | 0.007 | 4.76 | 0.000 | 0.019 | 0.045 |
| Remittance | 0.002 | 0.010 | 0.17 | 0.869 | -0.018 | 0.022 |
| Employment status | 0.250 | 0.182 | 1.38 | 0.168 | -0.106 | 0.606 |
| Livestock ownership | 0.004 | 0.021 | 0.2 | 0.840 | -0.038 | 0.046 |
| Cons | 0.025 | 0.036 | 0.69 | 0.490 | -0.045 | 0.094 |

presents various challenges, including higher living costs, limited access to agricultural land, and reliance on market-based food systems, all of which contribute to increased vulnerability to food insecurity. Studies by Ruel et al. [117], Crush and Frayne [118], and Battersby and Watson [119] have highlighted that urban populations may encounter difficulties in accessing affordable and nutritious food due to factors like higher food prices and limited availability of fresh produce. Additionally, income disparities and employment instability prevalent in urban settings can further exacerbate food insecurity. Lastly, Households in northern Ghana often have lower levels of food security than those in the south. The northern regions face unique challenges, including limited access to arable land, erratic rainfall patterns, and higher poverty rates. Furthermore, reliance on rain-fed agriculture in the north exposes households to climate variability and droughts, exacerbating food insecurity. Southern regions, on the other hand, benefit from more favourable agro-ecological conditions and better infrastructure, which leads to increased agricultural productivity and better market access [120,121].

## 4.4 The impact of water, energy, and food security on the well-being of the household

Table 7 presents the results of the impact of water, energy, and food security on households' well-being. Four models were estimated, with model 4 as the baseline model. In model 1, water security is negative and statistically significant at 1%, indicating that the probability of households reporting high well-being decreases with water security. Similarly, energy security is negative and significant at 1%, which implies that the probability of a household reporting higher well-being decreases with energy security as presented in model 2. Contrarily to water and energy security, food security is positive and significant for household well-being, which indicates that the probability of households reporting higher well-being increases with food security in model 3. These findings are consistent with the results in the baseline model (model 4). Austria Development Corporation [122] confirms that water, energy, and food security is defined as having access to clean drinking water and sanitation facilities, adequate energy, and sufficient and high-quality food essential for ensuring human well-being. This largely depends on the sustained availability of the resources. From the results, the unpopular and opposing assumptions about water and energy security may be related to stress, shortages of water due to physical and economic considerations, and insufficient energy supply to meet present demand, limiting the resources' possibility of enhancing household well-being. Also, a study by Adzawla, et al. [123] found that while access to water and sanitation facilities is important for households in Ghana, an increase in water security may not necessarily improve household well-being. The study suggests that factors such as income, education, and access to healthcare may have a greater impact on household well-being than access to water alone. Another study by Ren et al. [124] showed that an additional increase in energy security is less likely to improve household well-being in Ghana. The study found that despite increased access to energy services, particularly electricity, there has been limited progress in reducing energy poverty and improving living standards in Ghana due to several challenges, including inadequate institutional and regulatory frameworks, financial constraints, and limited capacity to absorb and utilize energy services.

Also, Felter and Robinson [125] admitted that prolonged water stress and scarcity lead to the spread of diseases such as cholera, typhoid, polio, hepatitis A, and diarrhoea, depriving them of their well-being. Furthermore, cooking, lighting, heating, cooling, cleaning, and technological, medical, and other life-sustaining devices [126,127] are examples of household energy uses. However, the inadequacy of energy supply is unlikely to improve their well-being. The recent food production programmes undertaken in the country could explain the positive

**Table 7. The impact of water, energy, and food security on the well-being of households.**

| VARIABLES | Model 1 | Model 2 | Model 3 | Model 4 |
|---|---|---|---|---|
| Water Security | -0.138*** | | | -0.121*** |
| | (0.0288) | | | (0.0290) |
| Energy Security | | -0.302*** | | -0.308*** |
| | | (0.0297) | | (0.0299) |
| Food Security | | | 0.216*** | 0.245*** |
| | | | (0.0356) | (0.0357) |
| Region | 0.900*** | 0.871*** | 0.893*** | 0.812*** |
| | (0.0466) | (0.0467) | (0.0466) | (0.0474) |
| Gender | 0.129** | 0.0719 | 0.122** | 0.0741 |
| | (0.0540) | (0.0543) | (0.0539) | (0.0544) |
| Age | 0.000629 | 0.00151 | 0.000453 | 0.00177 |
| | (0.00148) | (0.00149) | (0.00148) | (0.00150) |
| Household Size | -0.118*** | -0.120*** | -0.125*** | -0.129*** |
| | (0.00831) | (0.00834) | (0.00841) | (0.00846) |
| residence | 0.382*** | 0.305*** | 0.407*** | 0.342*** |
| | (0.0602) | (0.0607) | (0.0603) | (0.0610) |
| Marital Status | -0.0183 | -0.0161 | -0.0181 | -0.0188 |
| | (0.0146) | (0.0147) | (0.0146) | (0.0147) |
| Credit Access | 0.474*** | 0.415*** | 0.485*** | 0.410*** |
| | (0.0448) | (0.0453) | (0.0448) | (0.0454) |
| Remittance | 0.000131*** | 0.000117*** | 0.000134*** | 0.000117*** |
| | (0.000) | (0.000) | (0.000) | (0.000) |
| Household income | 0.00000136 | 0.00000159 | 0.00000132 | 0.00000155 |
| | (0.000) | (0.000) | (0.000) | (0.000) |
| Total Wage | 0.000129*** | 0.000111*** | 0.000141*** | 0.000108*** |
| | (0.000) | (0.000) | (0.000) | (0.000) |

*** $p < 0.01$,

** $p < 0.05$,

* $p < 0.1$.

Model 1(only water security inclusive), model 2(only energy security inclusive), model 3 (only food security inclusive), and model 4 (all securities are inclusive).

association of food security with household well-being. Furthermore, economic factors such as credit availability, remittances, and total wages were discovered to be more likely to improve household well-being. Furthermore, household characteristics such as region, gender, and residence location are more likely to improve household well-being, whereas household size was less likely to improve household well-being.

### 4.5 Diagnostics test

A diagnostic test was conducted to determine the robustness of the ordered Probit results by comparing the projected likelihood of the ordered Probit results to the actual means of household well-being. According to Table 8, the ordered Probit findings were fit for prediction.

## 5. Conclusions and policy implications

The study investigates the drivers of water, energy, and food security and its effect on household well-being in Ghana. Based on empirical analysis, several conclusions emerge. Firstly,

**Table 8. Actual means and predicted probabilities.**

| Variable | Obs | Mean | Std. Dev. | Min | Max |
|---|---|---|---|---|---|
| Well-being | | | | | |
| 1 | 2,735 | .366362 | .4818981 | 0 | 1 |
| 2 | 2,735 | .247532 | .4316573 | 0 | 1 |
| 3 | 2,735 | .1798903 | .3841663 | 0 | 1 |
| 4 | 2,735 | .1250457 | .3308312 | 0 | 1 |
| 5 | 2,735 | .08117 | .273146 | 0 | 1 |
| p1oprobit | 2,735 | .3634092 | .2336424 | 0 | .996003 |
| p2oprobit | 2,735 | .2528868 | .0612762 | 0 | .3163182 |
| p3oprobit | 2,735 | .1803968 | .0749177 | 8.60e-38 | .2638677 |
| p4oprobit | 2,735 | .1221604 | .0857065 | 6.51e-34 | .2787106 |
| p5oprobit | 2,735 | .0811468 | .1071429 | 6.03e-07 | 1 |

factors like age, credit access, household location, employment status, and livestock ownership contribute positively to household water security, whereas remittances, water supply management, water bills, and water quantity have negative impacts. Secondly, the study finds that age, marital status, household size, remittances, and livestock ownership significantly influence household energy security positively. Also, remittance and residence influence energy security negatively. Thirdly, marital status, household income, credit access, and household size positively impact household food security, whereas factors like residence and region of household location have a negative effect. Lastly, while water and energy security have a lesser influence on household well-being, food security is more likely to promote it.

However, the study suggests that policymakers revisit and implement the World Health Organization's water security initiative WASH (Water Sanitation and Hygiene) to improve the water security situation in Ghana. Policymakers should invest in the development and maintenance of water infrastructure, including dams, reservoirs, pipelines, and distribution networks, to ensure a reliable supply of clean water to households. Also, implementation of programs to improve sanitation facilities, such as toilets and wastewater treatment systems, to prevent water contamination and promote public health. Again, policymakers should implement policies to promote the use of innovative technologies for water treatment and purification to ensure the provision of safe and potable water. These measures are crucial for enhancing well-being.

The study suggests that government should invest in renewable energy sources. Ghana has abundant renewable energy sources such as solar, wind, and hydro, which can be harnessed to provide clean energy for households. By investing in renewable energy, the government can reduce the cost of energy consumption, improve energy security, and enhance household wellbeing. Also, the study suggests educating households on the importance of energy conservation and efficient energy use. The government should collaborate with civil society organizations to sensitize households on the need to conserve energy. Households can be taught how to use energy-efficient appliances, which will reduce the cost of energy consumption, improve energy security, and enhance household wellbeing. Also, the current food security programmes in Ghana should be maintained and intensified to increase the production capacity to cater for the growing food demand. Again, credit access strategies such as reduction of input prices, financial assistance, and interest rate payment policies should be regulated by policymakers and stakeholders to boost the production and accessibility of resources.

## Supporting information

**S1 Data.**
(XLSX)

## Acknowledgments

We thank discussants and participants at the Department of Economics Seminar series for their review comments and suggestions that has improved the quality of the manuscript. Any other errors remain the responsibilities of the authors.

## Author Contributions

**Conceptualization:** Foster Awindolla Asaki, Eric Fosu Oteng-Abayie.

**Data curation:** Foster Awindolla Asaki, Eric Fosu Oteng-Abayie, Franklin Bedakiyiba Baajike.

**Formal analysis:** Foster Awindolla Asaki, Eric Fosu Oteng-Abayie, Franklin Bedakiyiba Baajike.

**Investigation:** Foster Awindolla Asaki, Eric Fosu Oteng-Abayie.

**Methodology:** Foster Awindolla Asaki, Eric Fosu Oteng-Abayie.

**Project administration:** Foster Awindolla Asaki, Eric Fosu Oteng-Abayie.

**Resources:** Foster Awindolla Asaki, Eric Fosu Oteng-Abayie.

**Software:** Foster Awindolla Asaki, Eric Fosu Oteng-Abayie.

**Supervision:** Foster Awindolla Asaki, Eric Fosu Oteng-Abayie.

**Validation:** Foster Awindolla Asaki, Eric Fosu Oteng-Abayie.

**Visualization:** Foster Awindolla Asaki, Eric Fosu Oteng-Abayie.

**Writing – original draft:** Foster Awindolla Asaki, Eric Fosu Oteng-Abayie.

**Writing – review & editing:** Foster Awindolla Asaki, Eric Fosu Oteng-Abayie.

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
