## [Decision Letter · Decision Letter 0]

12 Dec 2023

PONE-D-23-27724Effects of Water, Energy, and Food Security on Household Well-beingPLOS ONE

Dear Dr. ASAKI,

Thank you for submitting your manuscript to PLOS ONE. After careful consideration, we feel that it has merit but does not fully meet PLOS ONE’s publication criteria as it currently stands. Therefore, we invite you to submit a revised version of the manuscript that addresses the points raised during the review process.

We look forward to receiving your revised manuscript.

Kind regards,

Fausto Cavallaro, PhD

Academic Editor

PLOS ONE

PONE-D-23-27724

Reviewers' comments:

Reviewer's Responses to Questions

**Comments to the Author**

1. Is the manuscript technically sound, and do the data support the conclusions?

Reviewer #1: Yes

Reviewer #2: Partly

Reviewer #3: No

2. Has the statistical analysis been performed appropriately and rigorously? 

Reviewer #1: Yes

Reviewer #2: No

Reviewer #3: Yes

3. Have the authors made all data underlying the findings in their manuscript fully available?

Reviewer #1: No

Reviewer #2: Yes

Reviewer #3: Yes

4. Is the manuscript presented in an intelligible fashion and written in standard English?

Reviewer #1: Yes

Reviewer #2: Yes

Reviewer #3: Yes

5. Review Comments to the Author

Reviewer #1: The research is interesting and contribute to the sustainable development goals. A study to examine the drivers of water, energy and food security and their effect on welfare is laudable and provide scientific evidence which inform policy. All the comments have been adequately addressed.

Reviewer #2: Comments to the authors

In a positive note, the authors have raised an important issue of concern in Africa, well-being affected by energy, food, and water security. These are issues of huge policy relevance. At its weakest, however, the article suffers from measurement and estimation problems as shown below.

1. WMS (Water supply systems) can indeed vary by household, even within the same locality. Factors such as infrastructure condition, household connections, and water usage patterns can lead to variations in water supply. However, in one village, water could be centrally managed, making this variable to be rather fixed for all households residing in that village. To deal with the fixed effects nature of the variable in a village where several sample households might have been considered for the analysis, there is a need to employ such statistical techniques as village fixed-effects regression to account for village-specific differences while estimating the effects of water management on water security.

2. Remittance and employment status can affect water security in multiple ways. Remittances can increase a household’s purchasing power, enabling them to invest in better water infrastructure or access alternative water sources. Similarly, employment status can influence water security as it affects a household’s income stability and ability to afford reliable water services. Nevertheless, the authors disregard this reality in their estimation.

3. Regarding livestock ownership (LSO), it is true that households with more livestock may have less water available for other purposes. This is because livestock require water for their own needs, and households often share their water supply with the animals. As a result, the water that could have been used for domestic or other productive purposes is allocated to livestock, potentially affecting overall water availability for the household. However, this is not considered in the regression model used.

4. The quantity of water a household uses per day (WQ) is indeed determined by various factors such as family size, distance from the water point, and water supply and management strategies. These factors can make the variable endogenous, meaning that it is influenced by other variables in the equation specified (see Equation 5). To account for endogeneity, researchers often employ instrumental variable approaches, such as using distance to the water point as an instrumental variable, to obtain unbiased estimates of the effects of other variables on water quantity. This is not taken into consideration in the article.

5. Energy security (ES) can be influenced by the number of livestock households own, as livestock rearing may require additional energy inputs for housing, feeding, and transportation. Education status can also affect energy security as it might be associated with income levels and access to modern energy sources. However, it is important to note that the specific mechanisms through which livestock ownership and education status affect energy security may vary and require further investigation. The authors need to take this issue into account.

6. Regarding the dummy variables WS, ES, and FS, without information on the cut-off values, it is challenging to interpret their exact meaning. Besides, while these variables are related, they are included in the household well-being equation (Equation 12) as separate indicators including almost similar variables affecting them. To address concerns about endogeneity and explore relationships among these variables, I suggest structural equation modeling, as it allows for the simultaneous examination of relationships between WS, ES and FS while estimating well-being. In fact, well-being is not measured in the article, and I wonder how the authors estimated it without measuring it.

Reviewer #3: Abstract

What is novel about the study? What problem exists?

Curious to know what is referred to well-being and how it was measured.

The objective in the abstract also refers to determining the impact. The analytical frameworks reported in the abstract do not relate to impact

Was the number of household used from the Ghana Living Standard Survey (GLSS) Wave 7 a sample or the total number of households. You need to state that. Also state the study design

Section 1.0

There is need for editing - some references are not in brackets (page 2 Line 56)

Rephrase page 2 Line 61 – it makes no sense “…shortage of …. demand...have worldwide ramifications….” How can shortages in demand have worldwide ramifications, or rather a shortage in supply

Rephrase page 9 Line 66

Page 4 Line 117-118, you criticize the use of primary data, yet your study uses secondary data which was collected through primary means. One might argue to the contrary, that primary data is better as it maintains internal validity and collected for its intended use

Page 5 Line 125, you justify use of longitudinal designs yet your study is not using longitudinal designs. It is actually using data collected through a cross sectional design since its data referring to just one wave.

Section 2.2

Page 8 Line 193-196, you need to rephrase, not sure what you were trying to say.

Your study can benefit from a conceptual framework showing the interrelations between water, energy and food as well as wellbeing

Page 9 Line 224, is this not a typo for capacity instead of capability

Section 3.2

How are your X and Y variables different when you state that Y represents independent variables (Page 14 Line 335-336). It is still not clear how the Y values were measured or conceptualized. How is the water, energy and food security measured

Page 15 Line 361, you refer to effect of water, energy and food on well-being, while in the abstract it refers to impact.

The referencing is also not consistent. Page 17 Line 410 you use Vancouver referencing but there is a different one in the text

Section 3.3

Still not clear how the PCA was used to come up with the water, energy and food indices

The indices in Table 1 indicate that the water, energy and food indices are actually binary as they are measured as dummy. How then do you account for your PCA. How did you account for collinearity when some of the variables used in Table 1 e.g. EB, but EB is also a subcomponent of ES

How was the GLSS conducted, you need to highlight how the survey was carried out by GLSS

Section 4.1

Page 21 Line 492, you refer to “males headed 1.25 of the households”, that is not shown in Table 2. Categorical variables such as urban or credit access are better reported in percentages and not their means. If you say 1.62 of the population have access to credit, it is also meaningless. Other things that are standing out in Table 2 is that for urban and credit access, you have minimum and maximum values of 0 and 1, respectively, but the means are more than the maximum values. How is that possible?

Section 4.2

Figure 1 is also not clear which bar represents more secure and which one represents less secure

Table 4 should be immediately placed after the paragraph of its first mention. It should be placed at Line 497

Section 4.4

Page 24 Line 537 – 540 The following statement is out of place “The empirical evidence of the Austria Development Corporation [110] confirms that water, energy, and food security is defined as having access to clean drinking water and sanitation facilities, adequate energy, and sufficient and high-quality food essential for ensuring human well-being.

Section 4.5

How did you come to the conclusion that the ordered probit was fit for prediction based on the values in Table 6

Section 5

No conclusions were reached. There was just a summarization of the results and recommendations

Throughout the whole document I think it’s prudent to change impact of water, energy and food security on well-being to effect of water, energy and food security on well-being

6. PLOS authors have the option to publish the peer review history of their article (what does this mean?). If published, this will include your full peer review and any attached files.

Reviewer #1: No

Reviewer #2: No

Reviewer #3: No

---

## [Author Response · Author response to Decision Letter 0]

23 Feb 2024

AUTHORS’ RESPONSE TO REVIEWERS' COMMENTS

We appreciate the editors’ and the reviewers' constructive comments that have further improved this manuscript. Based on your comments and other suggestions made, we provide the following responses.

Reviewers' comments

General comment:

In a positive note, the authors have raised an important issue of concern in Africa, well-being affected by energy, food, and water security. These are issues of huge policy relevance.

Response: We deeply appreciate the reviewer's positive assessment of our work and their recognition of the importance of the issues we address. It's encouraging to know that our research resonates with the significance of challenges related to energy, food, and water security in Ghana.

 Comment 1#: WMS (Water supply systems) can indeed vary by household, even within the same locality. Factors such as infrastructure condition, household connections, and water usage patterns can lead to variations in water supply. However, in one village, water could be centrally managed, making this variable to be rather fixed for all households residing in that village. To deal with the fixed effects nature of the variable in a village where several sample households might have been considered for the analysis, there is a need to employ such statistical techniques as village fixed-effects regression to account for village-specific differences while estimating the effects of water management on water security.

Response: Thank you for your insightful comment regarding the variability of water supply systems (WMS) within households and villages. We agree that factors such as infrastructure condition and household connections can lead to diverse water supply experiences, even within the same locality. We acknowledge that in some villages, water management may indeed be centrally controlled, resulting in a relatively fixed variable across households. To address the fixed effects nature of the variable, particularly in villages where multiple sample households are considered, statistical techniques such as village fixed-effects regression are often employed to account for village-specific differences. However, it's important to note that fixed effects models assume time-invariant effects (Verbeek, 2017), which may not hold in our case, given that the Ghana Living Standards Survey (GLSS) data represents a single point survey of households across the country. Despite the limitations of fixed effects models in this context, we have endeavored to capture the influence of water supply management on water security by measuring it as a dummy variable, with 1 indicating public supply management and 0 indicating private supply management. In our study, we use a binary variable to indicate whether the water supply is managed publicly or privately. While this approach does not account for the nuances of management within villages, it does allow us to evaluate the overall impact of various management systems on household water security.

Comment 2#: Remittance and employment status can affect water security in multiple ways. Remittances can increase a household’s purchasing power, enabling them to invest in better water infrastructure or access alternative water sources. Similarly, employment status can influence water security as it affects a household’s income stability and ability to afford reliable water services. Nevertheless, the authors disregard this reality in their estimation.

Response: We appreciate the reviewer's instructive comment regarding the impact of remittance and employment status on water security, and we acknowledge the oversight in our initial estimation. Recognizing the multifaceted nature of water security, we have integrated remittance and employment status variables into our model to more accurately capture their influence.

Comment 3#: Regarding livestock ownership (LSO), it is true that households with more livestock may have less water available for other purposes. This is because livestock require water for their own needs, and households often share their water supply with the animals. As a result, the water that could have been used for domestic or other productive purposes is allocated to livestock, potentially affecting overall water availability for the household. However, this is not considered in the regression model used.

Response: We appreciate the reviewer's useful observation about how livestock ownership may affect water availability in households. Recognizing the importance of this factor, we included livestock ownership (LSO) as a variable in our regression model to better understand its impact on overall water availability and usage patterns.

Comment 4#: 4. The quantity of water a household uses per day (WQ) is indeed determined by various factors such as family size, distance from the water point, and water supply and management strategies. These factors can make the variable endogenous, meaning that it is influenced by other variables in the equation specified (see Equation 5). To account for endogeneity, researchers often employ instrumental variable approaches, such as using distance to the water point as an instrumental variable, to obtain unbiased estimates of the effects of other variables on water quantity. This is not taken care of in the article.

Response: We appreciate the reviewer's astute observation regarding the potential endogeneity of the variable representing the quantity of water a household uses per day (WQ) in our model. Recognizing the importance of addressing this issue to ensure the validity of our estimates, we employed instrumental variable (IVPROBIT) methodology in our analysis. Incorporating instrumental variable approaches into our study, we aimed to mitigate potential endogeneity concerns and enhance the robustness of our findings. Thank you for highlighting this critical aspect.

Comment 5#: Energy security (ES) can be influenced by the number of livestock households own, as livestock rearing may require additional energy inputs for housing, feeding, and transportation. Education status can also affect energy security as it might be associated with income levels and access to modern energy sources. However, it is important to note that the specific mechanisms through which livestock ownership and education status affects energy security may vary and require further investigation. The authors need to take this issue into account.

Response: We appreciate the reviewer's instructive comment about the potential effects of livestock ownership and educational level on energy security. In our analysis, we did include livestock ownership and education status as variables in our model, recognizing their significance in determining household energy security.

Comment 6 #: Regarding the dummy variables WS, ES, and FS, without information on the cut-off values, it is challenging to interpret their exact meaning. Besides, while these variables are related, they are included in the household well-being equation (Equation 12) as separate indicators including almost similar variables affecting them. To address concerns about endogeneity and explore relationships among these variables, I suggest structural equation modeling, as it allows for the simultaneous examination of relationships between WS, ES and FS while estimating well-being. In fact, well-being is not measured in the article, and I wonder how the authors estimated it without measuring it. 

Response: We appreciate the detailed feedback provided by the reviewer regarding the interpretation of dummy variables WS, ES, and FS, as well as their inclusion in the household well-being equation. We acknowledge the importance of clarity regarding the cut-off values for these variables, and we apologize for any confusion. To clarify, in our study, WS, ES, and FS scores ranged from negative to positive values, with a cut-off value of zero. Values below zero were considered indicative of least security, while values above zero were deemed highly secure. We also recognize the interconnectedness of water security, energy security, and food security at the macro level, particularly in terms of their roles in energy generation and food preservation and production. However, at the household level, we treat WS, ES, and FS as independent variables due to their distinct impacts on household well-being. Regarding the measurement of well-being, we proxy it with quintile welfare, which is derived from the Ghana Living Standards Survey (GLSS) and ranges from 1 to 5, with 1 representing lower welfare and 5 indicating higher welfare. While we acknowledge the potential benefits of using structural equation modeling (SEM) to explore relationships among these variables, we believe that our current approach provides valuable insights into the determinants of household well-being within the context of water, energy, and food security. Thank you for your thoughtful suggestions, and we have ensured this clarification of these aspects in our methodology for improved understanding.

---

## [Decision Letter · Decision Letter 1]

28 May 2024

PONE-D-23-27724R1Effects of Water, Energy, and Food Security on Household Well-beingPLOS ONE

Dear Dr. ASAKI,

Thank you for submitting your manuscript to PLOS ONE. After careful consideration, we feel that it has merit but does not fully meet PLOS ONE’s publication criteria as it currently stands. Therefore, we invite you to submit a revised version of the manuscript that addresses the points raised during the review process.

We look forward to receiving your revised manuscript.

Kind regards,

Fausto Cavallaro, PhD

Academic Editor

PLOS ONE

Journal Requirements:

Additional Editor Comments:

The paper needs a further revision

Reviewers' comments:

Reviewer's Responses to Questions

**Comments to the Author**

1. If the authors have adequately addressed your comments raised in a previous round of review and you feel that this manuscript is now acceptable for publication, you may indicate that here to bypass the “Comments to the Author” section, enter your conflict of interest statement in the “Confidential to Editor” section, and submit your "Accept" recommendation.

Reviewer #2: All comments have been addressed

2. Is the manuscript technically sound, and do the data support the conclusions?

Reviewer #2: Yes

3. Has the statistical analysis been performed appropriately and rigorously? 

Reviewer #2: Yes

4. Have the authors made all data underlying the findings in their manuscript fully available?

Reviewer #2: Yes

5. Is the manuscript presented in an intelligible fashion and written in standard English?

Reviewer #2: Yes

6. Review Comments to the Author

Reviewer #2: Please refer to my previous comment #6, which reads as "To address concerns about endogeneity and explore relationships among these variables, I suggest structural equation modeling, as it allows for the simultaneous

examination of relationships between WS, ES and FS while estimating well-being. In fact, well-being is not measured in the article, and I wonder how the authors estimated it without measuring it." While the authors could resort to a different model (if they are not convinced about this), they should, at least treat the endogeneity of the variables in the model. Other than the three major ones (ES,FS and WS), income is also endogenous in the equations. I have not seen any attempt to deal with these issues. I am still of the opinion that authors highlight what they have done regarding this comment.

7. PLOS authors have the option to publish the peer review history of their article (what does this mean?). If published, this will include your full peer review and any attached files.

Reviewer #2: **Yes: **Bamlaku Alamirew Alemu

---

## [Author Response · Author response to Decision Letter 1]

31 May 2024

AUTHORS’ RESPONSE TO REVIEWERS' COMMENTS

We appreciate the editors’ and the reviewers' constructive comments that have further improved this manuscript. Based on your comments and other suggestions made, we provide the following responses.

Reviewer #2: Please refer to my previous comment #6, which reads as "To address concerns about endogeneity and explore relationships among these variables, I suggest structural equation modeling, as it allows for the simultaneous examination of relationships between WS, ES and FS while estimating well-being. In fact, well-being is not measured in the article, and I wonder how the authors estimated it without measuring it." While the authors could resort to a different model (if they are not convinced about this), they should, at least treat the endogeneity of the variables in the model. Other than the three major ones (ES, FS and WS), income is also endogenous in the equations. I have not seen any attempt to deal with these issues. I am still of the opinion that authors highlight what they have done regarding this comment.

Response: Thanks to the reviewer for pointing out this. While it is true that there is the possibility of the presence of endogeneity, Angrist and Pischke (2008) argue that instrumental variable can be allowed since this is a large data set. They argue that in large datasets, the sheer volume of data can provide enough variability to identify causal relationships even without the use of instrumental variables. Again, Wooldridge (2010) noted that in large datasets, the presence of numerous observations and variables can provide enough information to account for omitted variable bias. The dataset used for this study is 2, 735 observations, thus large enough to identify causal relationships even without the use of instrumental variables. Regarding well-being measurement, this has been addressed in the study. See line 196 to line 201.

---

## [Editor Report · Decision Letter 2]

28 Jun 2024

Effects of Water, Energy, and Food Security on Household Well-being

PONE-D-23-27724R2

Dear Dr. ASAKI,

We’re pleased to inform you that your manuscript has been judged scientifically suitable for publication and will be formally accepted for publication once it meets all outstanding technical requirements.

Kind regards,

Fausto Cavallaro, PhD

Academic Editor

PLOS ONE

---

## [Editor Report · Acceptance letter]

3 Jul 2024

PONE-D-23-27724R2 

PLOS ONE

Dear Dr. Asaki, 

I'm pleased to inform you that your manuscript has been deemed suitable for publication in PLOS ONE. Congratulations! Your manuscript is now being handed over to our production team.

Kind regards, 

on behalf of

Professor Fausto Cavallaro 

Academic Editor

PLOS ONE